# Hydrochemical Characteristics and Groundwater Quality Assessment in the Diluvial Fan of Gaoqiao, Emei Mountain, China

**Yanna Yang [1], Wenlai Xu [1,2,3,*], Jinyao Chen [1,*], Qiang Chen [1] and Zhicheng Pan [2]**

1 State Key Laboratory of Geohazard Prevention and Geoenvironment Protection, Chengdu University of Technology, Chengdu 610059, China; yangyanna@cdut.cn (Y.Y.); chenqiang200301081@163.com (Q.C.)
2 Haitian Water Grp Co Ltd., Chengdu 610059, China; pan12487616@126.com
3 Department of Chemical Engineering, Tokyo University of Agriculture & Technology, Tokyo 1848588, Japan
* Correspondence: xuwenlai2012@cdut.cn (W.X.); txgsfy@163.com (J.C.); Tel.: +86-028-84078874 (W.X.)

**Abstract:** Methods for groundwater quality and pollution assessment are applied extensively, but it is difficult to determine a unified evaluation model. On the basis of hydrogeochemical characteristics analysis in 2016 compared with that in 1995, the five-element connection number SPA (set pair analysis) method was applied to evaluate the groundwater quality of the Gaoqiao diluvial fan under the influence of hydrogeological conditions and human activities along the flow path in our work. Descriptive statistics methods, Piper diagram, and a Schoeller diagram were also used to analyze the hydrochemical characteristics of groundwater such as chemical components, total dissolved solid, and total hardness. The change of the typical pollutant of fluorine was analyzed to evaluate the groundwater quality under the influence of human activities. The results showed that the groundwater quality in the study area was more in rank $\Pi$. The basic hydrochemical types of shallow groundwater were $HCO_3$-Ca·Mg and $HCO_3$·$SO_4$-Ca·Mg. The influencing factors of the hydrochemical component of groundwater were identified in the Gaoqiao diluvial fan. The quality of groundwater changed slightly from the top to the edge of the fan due to the water–rock interaction except for in Yucun and Hucun influenced by human activities. The assessment result can provide a scientific basis for the pollution prevention and changing process control of the groundwater in the hydrogeological unit of the Gaoqiao diluvial fan.

**Keywords:** Gaoqiao diluvial fan; groundwater quality assessment method; groundwater resources; hydrochemical characteristics; the five-element connection number SPA method

## 1. Introduction

Chinese people's spare time is increasingly including on tourism with the development of the economy and urban progress. The development of the tourism industry has placed a burden on the demands of groundwater due to the increasing population [1–3]. With the development of tourism and the increase in visitors, the development and security of water resources are becoming key issues. The quality of groundwater is as important as its quantity because it is key in determining its suitability for drinking, domestic, irrigation, and industrial purposes [4]. It is of great significance for the sustainable utilization of groundwater resources to find out the current situation of groundwater quality and recovery degree and to carry out pollution control and remediation measures as soon as possible.

Methods for groundwater quality and pollution assessment were applied extensively [5]; however, no firmly established assessment method has been accepted and applied extensively [6]. The "grey water footprint" (GWF) was used to assess the sustainability of pollution produced by human activities

and to measure the amount of water required to assimilate a polluting load produced from anthropic activity [7]. The Driver-Pressure-State-Impact-Response (DPSIR) framework provided a simplified description of the various components of a complex environmental system to obtain information useful for the management of water resources [8], applied in a central area of the Salento peninsula to identify environmental and human factors influencing the quality of groundwater [9].

Due to the complexity of groundwater pollution, the assessment processes are divided into a single factor evaluation and comprehensive factor evaluation. Comprehensive factor evaluation methods are used widely such as the comprehensive index method, fuzzy mathematics method, grey clustering method, and artificial neural network method [10–15]. Mehra [16] evaluated groundwater quality with the weighted index method and proposed an integrated assessment of groundwater for agricultural use using a geographical information system (GIS), resulting in some important and useful results that decision makers could apply to measures supporting sustainable groundwater resource management in the study area. Liu [17] applied a factor analysis method revealing that the areas of high seawater salinization and arsenic pollution corresponded with underground water overpumped in Taiwan. N. Subba Rao [1] analyzed the hydrochemical characteristics and controlling factors of Andhra Pradesh groundwater by using a Piper diagram, correlation analysis, and other methods. Ramin Sarikhani [2] used descriptive statistics, Piper diagram, Schoeller diagram, and correlation analysis methods to analyze the geochemical characteristics of groundwater in Bushehr Province and its relationship with regional groundwater quality.

An appropriate assessment method of groundwater quality contributes significantly to understanding the changing process of the chemical composition of the region [18]. The result of assessment can better reflect the changing process of groundwater quality, and be utilized to evaluate the quality of groundwater and guarantee its safety [19,20]. Huan [21] pointed out that the interaction degree between groundwater and rock media in different depths and regions in the fan sector was different in the study area of the alluvial–diluvial fan of Yongding River. Guo [22] pointed out that the groundwater hydrochemistry field had the characteristics of stratified zoning in the alluvial–diluvial fan of Chaobai River. Yasong Li [23] pointed out that the hydrological environment and human activities resulted in the introduction of toxic metals, nitrogen, and organic substances into the groundwater in the study of an alluvial fan area of the Hunhe River. Wu et al. [24,25] studied the characteristics of groundwater and evaluated the water quality in the Gaoqiao diluvial fan area in 1995. The results showed that most areas were polluted lightly or not polluted; Futian, Yangcun, and Huaiyusi were polluted to some extent; and Sifangbei, Wenchanggong, and Luomuzhen were polluted heavily.

It is difficult to determine a unified evaluation model due to the uncertainty, fuzziness, and greyness characteristics of groundwater pollution [6]. Set pair analysis (SPA) is a method known as connection mathematics and used to tackle the interaction between systematic certainty and uncertainty. Based on the theory of identity-discrepant-contrary (IDC), SPA solves the certainty or uncertainty problems by quantitative analysis. It is superior to the uncertainties since it is not necessary to distinguish the information definitely and has obvious advantages in dealing with fuzzy, random, intermediate, and incomplete information. It has been applied in the evaluation of impacts of groundwater quality assessment [6], water resources management problems [26–28], and other groundwater analysis issues.

The historical chemistry of groundwater provides us with a datum against which changes in chemical composition caused by pollution can be measured. Historical water quality data are one of the best and most direct approaches to determine the background concentration of a contaminant [6]. On the basis of hydrogeochemical components analysis in 2016 compared with that in 1995, the five-element connection number SPA method was proposed and applied to analyze and evaluate the groundwater quality of the Gaoqiao diluvial fan under the influence of human activities in this work.

## 2. Materials and Methods

### 2.1. Study Area

Mount Emei is located in Leshan City, Sichuan Province, China. It is one of the four famous Buddhist Mountains and is a Chinese national key cultural relics' protection unit, national key scenic spots, and has been listed on the UNESCO World Cultural and Natural Heritage List since 1996. It is a typical block mountain formed by the Himalayan movement and the accompanying uplift of the Qinghai–Tibet plateau. The elevation of the western mountain area is over 3000 m above sea level. The annual average rainfall in the mountainous area is 1786 mm.

The study area of the Gaoqiao diluvial fan is located at the eastern foot of Mount Emei, as shown in Figure 1. It is 3 km away from the top of Mount Emei, and the elevation on average is 450 m above sea level, belonging to the prospective reserve zone of Mount Emei. Luomuzhen is a secondary scenic spot located in the middle of the Gaoqiao diluvial fan.

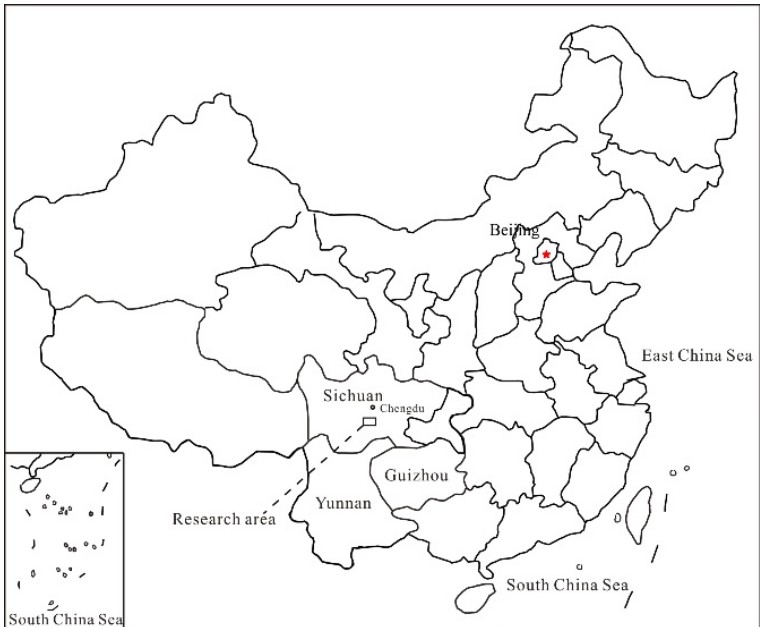

**Figure 1.** The position of the study area is located in the southeastern foot of Mount Emei, Sichuan Province, China.

The study area is approximately 16 km$^2$ and the longitudinal length NW–SE extending is about 7 km. The Quaternary stratum ($Q_4$, $Q_3^3$, $Q_3^2$, $Q_3^1$, etc.) is a widely horizontal distribution. The deposits in the study area are mainly basalt, sandstone, and dolomite, with sand and clay intermixed. The thickness of the accumulation layer is large, and the stratum separation in the vertical direction is poor. The rich groundwater of the study area comes from rainfall and lateral recharge from the mountainous area and discharges from the middle and edge of the fan with wells and overflow springs. The average shallow groundwater level in the study area is shown in Figure 2.

From the top to the edge, the fan is divided into three parts according to the gravel particle size of the material composition. In the top area, the gravel particles were typically about 20–50 cm in diameter; in the middle area, the size was much smaller such as gravel, sand, and silt; and in the edge area, the smallest compositions were found such as sub-sandy soil, and sub-clay. The material compositions of the three parts of the fan are shown in Figure 3.

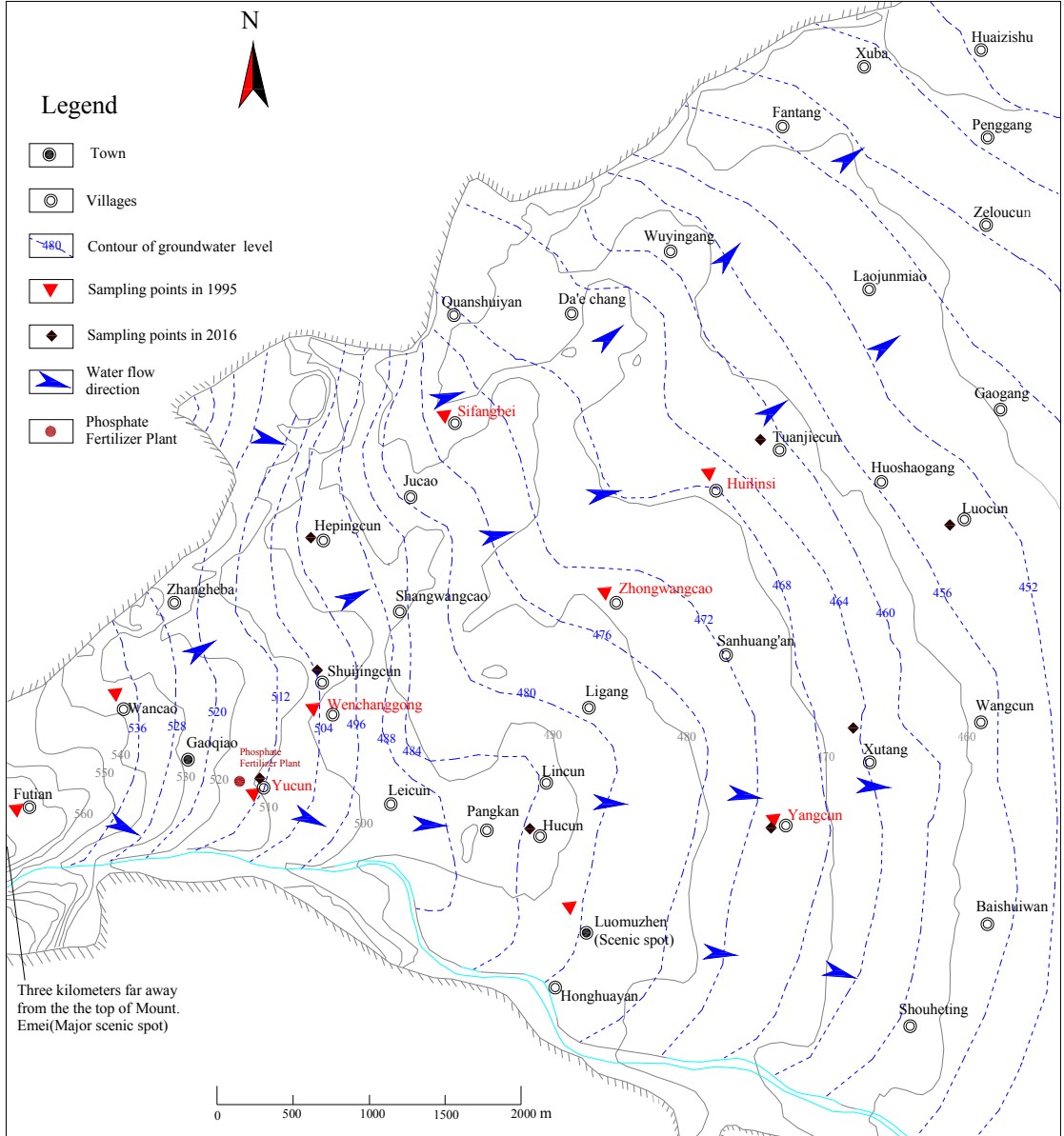

**Figure 2.** The groundwater flow, sites of pollution sources, and hydrochemical sampling points in the study area.

The groundwater of the Gaoqiao diluvial fan provides a water source for industrial, agricultural, and residential use. Before 1990, there was a phosphate fertilizer plant, lime mine plant, and cement plant, forming industrial pollution sources, which resulted in the deterioration of groundwater quality [3]. The phosphate fertilizer plant had produced calcium, magnesium, phosphorus, and potassium fertilizer from phosphorite ore with dolomite as a cosolvent, coking coal, and white coal by the blast furnace method since 1964. The annual output had increased from 20,000 tons in 1964 annually to 130,000 tons in 1984. Due to the high fluorine content of the raw materials and fuels, the backwardness of the production technology, a low chimney, and the absence of exhaust gas and water recovery equipment, a large amount of fluoride was discharged into the ecological and groundwater environment during the production process from the plant. The content of fluorine in the wastewater, measured by the environmental protection department in 1978, was as high as 45 mg/L. Coercive measures had been taken to eliminate fluorine in the wastewater of the plant from 1980 to 1996. However, the "dental fluorosis" of the villagers in Yucun, 100 m downstream of the plant, was still

showing heavy pollution of fluorine in the groundwater from 1980s. All the industrial pollution sources had been banned since 1996 and the groundwater environment was slowly recovering.

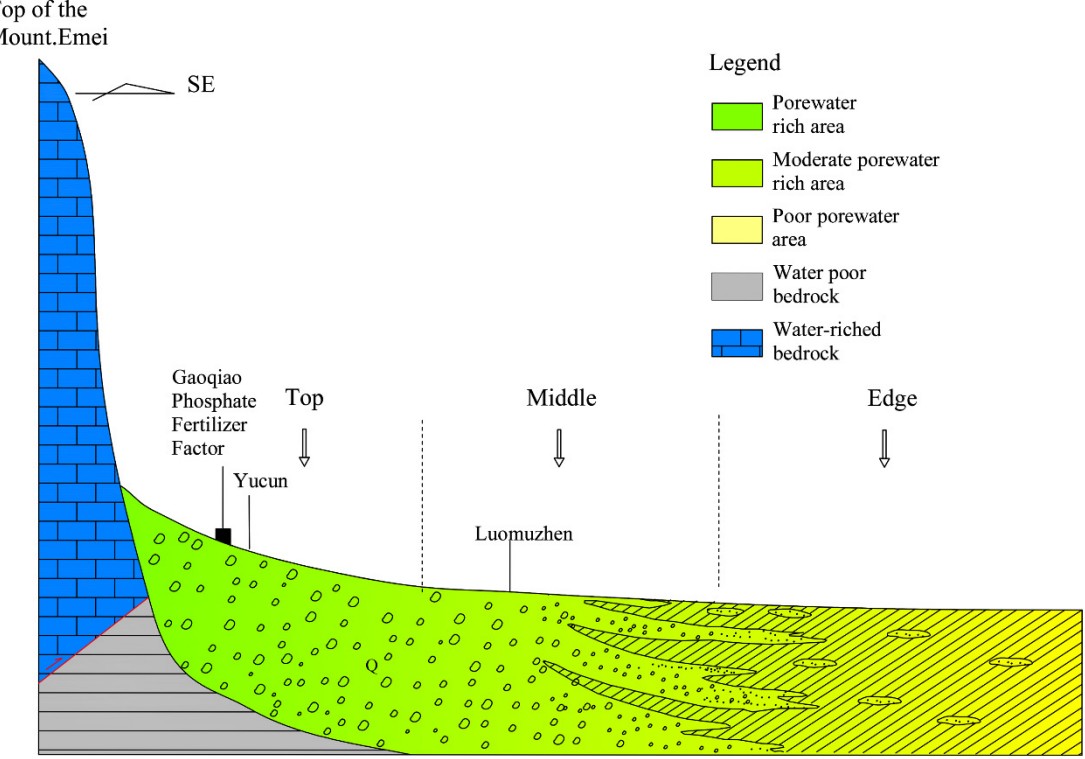

**Figure 3.** The hydrogeological profile figure of the composition features of the Gaoqiao diluvial fan.

## 2.2. Sample Collection and Analysis

The historical water quality data of hydrochemical components in 1995 was obtained from paper [24]. A total of eight groups of water samples were collected from the shallow groundwater of the fan in August 2016. The points were sampled at a similar area of the fan between 1995 and 2016, shown in Figure 2. The sampling points were distributed in the vertical and horizontal sections from the recharge area to the discharge area of the Gaoqiao diluvial fan according to the flow direction, as shown in Figure 3. The hydrological characteristics of sampling points were shown in Table 1. During the process of filtering samples, blank samples (deionized water), and parallel samples were taken into consideration to control the reliability.

Water samples were tested by SiChuan Tianshengyuan Environmental Limited Corporation. The test method used to detect the presence of $K^+$, Na, $Ca^{2+}$, $Mg^{2+}$, and Fe was inductively coupled plasma atomic emission spectrometry; the method used to detect F, $Cl^-$, and $SO_4^{2-}$ was ion chromatography; and the method used to detect $HCO_3^-$ was the acid–base titration. The measurement method used to obtain the total dissolved solid was a weighing method. The total hardness (measured by $CaCO_3$) was titrated with an EDTA (Edta direct titration method) titration method, and the test method for ammonia nitrogen (measured by N) was nashi reagent spectrophotometry.

**Table 1.** The hydrological characteristics of the sampling points in the Gaoqiao diluvial fan.

| Sampling Time | Site | Geographic Location | Characteristics of Samples |
|---|---|---|---|
| 1995 | Wancao | Top | Spring |
| | Futian | | Spring |
| | Yucun | | Well |
| | Wenchanggong | | Spring |
| | Sifangbei | | Well |
| | Luouzhen | Middle | Well |
| | Zhongwangcao | | Well |
| | Yangcun | | Well |
| | Huilingsi | Edge | Well |
| 2016 | Yucun | Top | Spring |
| | Shuijincun | | Well |
| | Hepingcun | | Well |
| | Hucun | Middle | Well |
| | Yangcun | | Well |
| | Xutangcun | Edge | Spring |
| | Tuanjiecun | | Well |
| | Luocun | | Well |

*2.3. The Principle of Five-Element Connection Number SPA in Groundwater Assessment*

Set pair analysis (SPA) is used to deal with the problem of determination uncertainty. The essence of this modified uncertainty theory is to treat the certainties and uncertainties as an integrated system. The treatment can be implemented by studying the relationship between the certainty and the uncertainty of an object, which is relative with three aspects of identity-discrepant-contrary (IDC) [29–31]. The method proceeds as follows: (1) set pairs of two relative sets in uncertain systems are established; (2) attributes are analyzed and calculated using identity, discrepancy, and contrary; (3) the degree of connection of a pair of combinations can be established according to the different attributes. In SPA, identity, discrepancy, and contrary are transformed under certain conditions, and the evaluation of groundwater environmental quality can be realized through this relationship between interconnection and restriction [29].

To analyze the characteristics of the two sets (A and B) under a certain background, as well as the similarities and differences, the relationship and the connection degree between A and B is defined as the following:

$$L = a + bi + cj \tag{1}$$

The discrepancy $bi$ in Equation (1) is rewritten as $b_1i_1 + b_2i_2 + \ldots + b_ni_n$ with people's deeper understanding of the associative coefficient in set pair analysis theory, getting the function with hierarchical structure, i.e., multiple connection number. When $n = 3$, it is a five-element connection number ($L = a + bi + cj + dk + el$). The associative components ($a, b, c, d, e$) have an optimal order, that is, the things represented by $a$ are better than those represented by $b$, and so on. When the value of $i, j, k$, and $l$ are not considered, they are only used as markers. When these four values need to be considered, they have both a gain and a decay effect on $a$. This fully reflects the relationship between the number of connections and the unity of opposites between the components. The method is called five-element connection number SPA.

Here, the hydrochemical characteristics and the complexity of the quality grade of shallow groundwater were studied using a five-element connection number of SPA. The index content and groundwater quality standard of samples were taken as a set pair in groundwater quality evaluation when proper parameters were selected as evaluation indexes. According to the *Groundwater Quality Standards of China* (GB/14848-2017) [32], groundwater quality can be classified into five ranks. Rank III is always chosen as the drinking water quality standard in the groundwater quality evaluation.

The rank values from rank I (excellent quality water) to rank V (extremely poor water) are divided by $S$, $F^1$, $F^2$, $F^3$, $P$. So, the connection degree is expressed as Equation (2):

$$\mu = \frac{S}{N} + \frac{F^1}{N}i + \frac{F^2}{N}j + \frac{F^3}{N}k + \frac{P}{N}l, \tag{2}$$

where, $N$ is evaluation value; $\mu$ is the identity degree coefficient; $i$, $j$, and $k$ are coefficients of discrepancy degree; $l$ is the coefficient of contrary degree.

If $a = S/N$, $b = F1/N$, $c = F2/N$, $d = F3/N$, $e = P/N$, that is, $a$ and $e$ respect identity and contrary degree and $b$, $c$, and $d$ are discrepancy degree. Therefore, Equation (2) can be simplified as Equation (3):

$$\mu = a + bi + cj + dk + el, \tag{3}$$

where, the identity degree ($a$) and the contrary degree ($e$) are relatively determined, while the discrepancy degree ($b$, $c$, $d$) is relatively undetermined. $a$, $b$, $c$, $d$, $e$ and $i$, $j$, $k$, $l$, and other parameters are interrelated, interacted, and restricted. The higher the value of $a$ or the lower the value of $e$ is, the better the quality of groundwater is.

By analyzing the quantitative relationship between the evaluation index and the groundwater quality grade of each sample, it can be seen that the groundwater quality is at the same level. However, the water environment is different due to a different evaluation index. There are four limiting values between the rank I and rank V water quality indicators, which are considered to be the basis of identity, discrepancy, and contrary values. Therefore, the connection degree plays a key role in the comprehensive assessment of water quality based on SPA, which can be calculated by Equation (4).

$$\mu = \begin{cases} 1 + 0i + 0j + 0k + 0l, & x \in [0, \ S_1] \\ \dfrac{S_2 - x}{S_2 - S_1} + \dfrac{x - S_2}{S_2 - S_1}i + 0j + 0k + 0l, & x \in (S_1, \ S_2] \\ 0 + \dfrac{S_3 - x}{S_3 - S_2}i + \dfrac{x - S_2}{S_3 - S_2}j + 0k + 0l, & x \in (S_2, \ S_3], \\ 0 + 0i + \dfrac{S_4 - x}{S_4 - S_3}j + \dfrac{x - S_3}{S_4 - S_3}k + 0l, & x \in (S_3, \ S_4] \\ 0 + 0i + 0j + 0k + 1l, & x \in [S_4, \ +\infty) \end{cases} \tag{4}$$

where, $S1$, $S2$, $S3$, and $S4$ are the limiting values of water quality rank I, II, III, and IV; $x$ is the measured value of water quality status; $m$ is the $m$-th groundwater sampling point; $p$ is the evaluation index.

According to the calculation results of Equation (3), the average value of each is taken to obtain the average contact degree ($\mu m$) of the evaluation sample ($m$). The specific calculation Equation (5) is as follows:

$$\mu_m = \frac{1}{n} \sum_{p=1}^{n} \mu_{mp} \quad 1 \leq p \leq n, \tag{5}$$

The correlation degree of different evaluation ranks to the groundwater quality of sampling water in the study area can be obtained according to Equations (4) and (5), and the grade I of groundwater quality can be evaluated. Therefore, the water quality composite grade of the $m$ sample is:

$$Gm = 1 \times a + 2 \times b + 3 \times c + 4 \times d + 5 \times e, \tag{6}$$

where $G_m$ is the complexity of the quality grade of groundwater.

There were eight samples and eight indicators selected as evaluation indexes to evaluate the groundwater environmental quality in the study area by five-element connection number of SPA. The indexes were total hardness (TH), total dissolved solids (TDS), sulfate ($SO_4^{2-}$), chloride ($Cl^-$), nitrate ($NO_3^-$), fluoride (F), ammonia nitrogen ($NH_4^+$), and iron (Fe), as shown in Table 2.

**Table 2.** The value of indexes in the Groundwater Quality Standards of China (GB/14848-2017) (mg/L).

| Indexes | I | II | III | IV | V |
|---------|-----|------|------|------|-------|
| TH | $\leq 150$ | $\leq 300$ | $\leq 450$ | $\leq 650$ | $>650$ |
| TDS | $\leq 300$ | $\leq 500$ | $\leq 1000$ | $\leq 2000$ | $>2000$ |
| $SO_4^{2-}$ | $\leq 50$ | $\leq 150$ | $\leq 250$ | $\leq 350$ | $350$ |
| $Cl^-$ | $\leq 50$ | $\leq 150$ | $\leq 250$ | $\leq 350$ | $>350$ |
| $NO_3^-$ | $\leq 2$ | $\leq 5$ | $\leq 20$ | $\leq 30$ | $>30$ |
| F | $\leq 1$ | $\leq 1$ | $\leq 1$ | $\leq 2$ | $>2$ |
| $NH_4^+$ | $\leq 0.02$ | $\leq 0.1$ | $\leq 0.5$ | $\leq 1.5$ | $>1.5$ |
| Fe | $\leq 0.1$ | $\leq 0.2$ | $\leq 0.3$ | $\leq 2$ | $>2$ |

## 3. Result and Discussion

### 3.1. Hydrochemical Characteristics of Groundwater

#### 3.1.1. Chemical Components of Groundwater

The statistics regarding the chemical parameters and background parameters of each sample are shown in Table 3.

**Table 3.** The main ion value of the water samples in the study area (mg/L).

| Position | Time | site | $K^+ + Na^+$ | $Ca^{2+}$ | $Mg^{2+}$ | $Cl^-$ | $SO_4^{2-}$ | $HCO_3^-$ | $CO_3^{2-}$ | $NO_3^-$ | TH | TDS | F | $NH_4^+$ | Fe |
|----------|------|------|------|------|------|------|------|------|------|------|------|------|------|------|------|
| Top | 1995 | Wancao | 7.06 | 79.26 | 24.05 | 9.93 | 65.32 | 285.4 | 0.00 | / | 296.90 | 328.74 | 0.10 | / | / |
| | | Futian | 16.38 | 56.97 | 22.05 | 12.10 | 42.27 | 252.18 | 0.00 | / | 233.00 | 276.55 | 0.50 | / | / |
| | | Yucun | 9.38 | 79.10 | 26.65 | 10.60 | 82.61 | 239.88 | 0.00 | / | 307.20 | 329.38 | 1.00 | / | / |
| | | Wenchanggong | 7.55 | 76.78 | 22.54 | 8.51 | 59.56 | 282.94 | 0.00 | / | 269.50 | 318.51 | 1.10 | / | / |
| | 2016 | Yucun | 23.9 | 73.10 | 24.90 | 5.50 | 69.10 | 265.00 | 0.00 | 23.00 | 285.00 | 364.00 | 2.86 | <0.02 | <0.05 |
| | | Shuijincun | 10.20 | 51.10 | 41.90 | 3.98 | 33.40 | 302.00 | 0.00 | 28.60 | 300.00 | 330.00 | 0.95 | 0.05 | <0.05 |
| | | Hepingcun | 30.30 | 53.10 | 24.90 | 6.47 | 62.90 | 219.00 | 0.00 | 32.30 | 235.00 | 332.00 | 0.34 | <0.02 | <0.05 |
| Middle | 1995 | Sifangbei | 11.29 | 81.74 | 27.55 | 37.6 | 42.27 | 227.58 | 0.00 | / | 317.50 | 351.54 | 0.22 | / | / |
| | | Luomuzhen | 20.31 | 102.40 | 43.58 | 22.00 | 90.27 | 394.88 | 0.00 | / | 435.00 | 478.10 | 0.34 | / | / |
| | | Zhongwangcao | 12.67 | 81.74 | 32.56 | 11.30 | 42.27 | 356.75 | 0.00 | / | 338.10 | 359.78 | 0.70 | / | / |
| | | Yangcun | 12.28 | 97.43 | 26.05 | 15.60 | 74.54 | 332.14 | 0.00 | / | 350.50 | 392.59 | 0.22 | / | / |
| | 2016 | Hucun | 63.60 | 120.00 | 45.00 | 152.00 | 95.60 | 320.00 | 0.00 | 67.40 | 485.00 | 716.00 | 0.29 | <0.02 | <0.05 |
| | | Yangcun | 18.00 | 74.10 | 43.20 | 8.94 | 86.10 | 323.00 | 0.00 | 32.70 | 363.00 | 435.00 | 0.31 | <0.02 | <0.05 |
| Edge | 1995 | Huilingsi | 24.75 | 88.34 | 32.56 | 15.60 | 78.77 | 350.60 | 0.00 | / | 354.60 | 416.22 | 0.50 | / | / |
| | 2016 | Xutangcun | 12.20 | 89.10 | 33.44 | 7.99 | 71.70 | 308.00 | 0.00 | 59.60 | 360.00 | 438.00 | 0.30 | <0.02 | <0.05 |
| | | Tuanjiecun | 17.00 | 72.10 | 45.60 | 15.60 | 119.00 | 183.00 | 0.00 | 124.00 | 368.00 | 494.00 | 0.24 | <0.02 | <0.05 |
| | | Luocun | 22.40 | 104.00 | 39.50 | 15.00 | 72.80 | 320.00 | 0.00 | 112.00 | 423 | 537 | 0.33 | <0.02 | <0.05 |

It can be seen from Table 3 that from the top to the edge of the fan, the main cation in the constant components of groundwater is $Ca^{2+}$ in the study area, except Hucun in 2016. The main anion in the constant components of groundwater is $HCO_3^-$. The chemical components of groundwater were a reflection of migration law conforming to the rapid recharge and rapid discharge of shallow groundwater in the fan.

#### 3.1.2. The Hydrochemical Characteristics of Groundwater

The geochemical evolution of groundwater can be explained by the concentration of the main cation and anion in the Piper diagram. According to the distribution, the concentration relationship between the cation and anion in groundwater is evident, and the hydrogeochemistry of the groundwater is revealed. According to the Piper diagrams in Figures 4 and 5, in the study area, the main cation and anion of the groundwater were concentrated in Zone 5, where carbonate hardness exceeds 50%; cations were mainly $Ca^{2+}$, and anions were mainly $HCO_3^-$ and $SO_4^{2-}$. From the top to the edge of the fan, the concentration of cation and anion gradually increased, and the proportion of each ion changed slightly. In some areas, the cation changed from $Ca^{2+}$ to $Mg^{2+}$ and $Na^+$, and the anion changed from $HCO_3^-$ and $SO_4^{2-}$ to $Cl^-$. The basic hydrochemical types of shallow groundwater were $HCO_3$-Ca·Mg and $HCO_3$·$SO_4$-Ca·Mg. The gradual change of the main cation and

anion of the groundwater was attributed to the water–rock interaction along the flow path according to the mineralogy of aquifers. The hydrochemical type of in Yucun was $HCO_3 \cdot Cl\text{-}Ca \cdot Mg$, which was affected by the phosphate fertilizer plant.

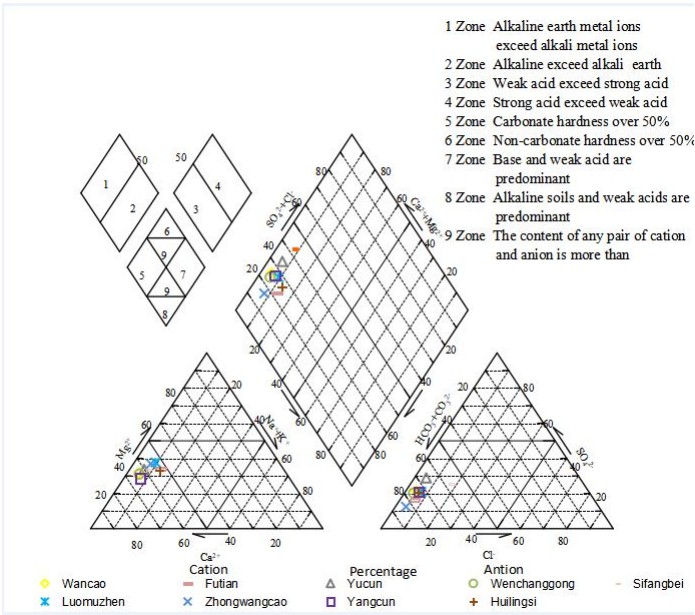

**Figure 4.** The main cation and anion of groundwater in 1995 in the study area shown in a piper diagram.

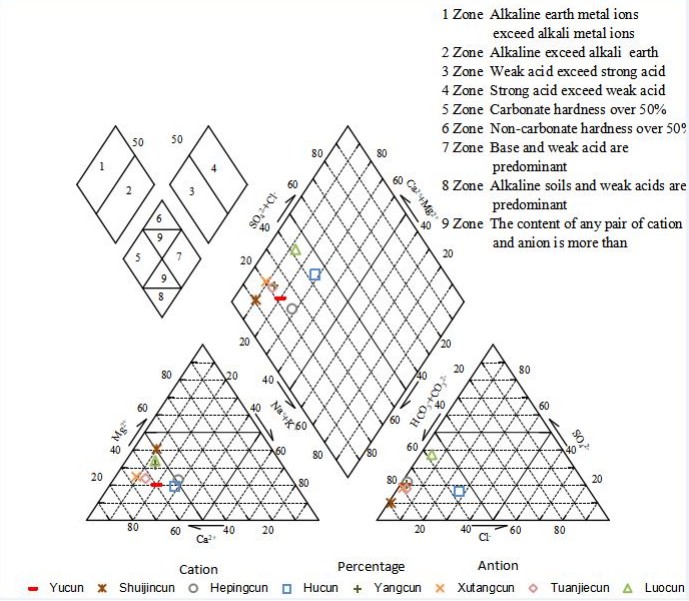

**Figure 5.** The main cation and anion of groundwater in 2016 in the study area shown in a Piper diagram.

### 3.1.3. Total Dissolved Solid Characteristics

From the top to the edge of the fan, the total dissolved solid (TDS) of the groundwater varied from 276.55 mg/L to 478.1 mg/L, with an average value of 361.27 mg/L in 1995, and it varied from 330 mg/L to 716 mg/L, with an average value of 455.78 mg/L in 2016. The total dissolved solids in 2016 increased by 30% compared with 1995. The slight increase of TDS ions was attributed to the water–rock interaction along the flow path.

### 3.1.4. Total Hardness (TH) Characteristics

The total hardness (TH) is measured by the value of calcium and magnesium, denoted as calcium carbonate. In 1995, the total hardness of groundwater in the study area varied from 269.53 mg/L to 354.58 mg/L, with an average value of 322.48 mg/L, and it varied from 235 mg/L to 485 mg/L, with an average value of 352.38 mg/L in 2016. The hardness of groundwater had not greatly changed from 1995 to 2016. The hardness of the groundwater in the study area was at least 150 mg/L, which belonged to fairly hard water. Some samples were of hard water, and one sample was more than 450 mg/L, which belonged to very hard water, as shown in Table 4.

**Table 4.** Classification of groundwater hardness in the study area (mg/L)

| TH (mg/L) | Groundwater Soft Hard Type | Samples | |
|---|---|---|---|
| | | In 1995 | In 2016 |
| 0~75 | Very soft water | 0 | 0 |
| 75~150 | Soft water | 0 | 0 |
| 150~300 | Fairly hard water | 3 | 3 |
| 300~450 | Hard water | 6 | 4 |
| >450 | Very hard water | 0 | 1 |

### 3.1.5. Characteristics of Pollutant Fluorine in Groundwater

Yucun is located 100 m downstream of the Gaoqiao phosphate fertilizer plant. According to the historical water quality data, the content of fluorine in the groundwater of Yucun was 1.1 mg/L in 1995. The content of fluorine was 2.86 mg/L in the groundwater of Yucun in 2016. Although measures had been taken from 1980 to 1996 to eliminate the fluorine in the wastewater of the phosphate fertilizer plant, the fluorine had been enriched in the soil and vegetation around the plant from the waste gas, and the residues of pollutants were constantly released after the plant was shut down.

### 3.1.6. Characteristics of a Schoeller Diagram

A Schoeller diagram is a semilog of the concentration of the main ion components of water, showing the different hydrogeochemical water types of the main ions on the same chart. Solute concentrations are plotted as a Schoeller diagram to visualize the relative changes in solute concentrations in the sample, which allows the similarity of the ratios between solutes to be intuitively assessed. In a Schoeller diagram, the slope of the line is associated with the concentration of the solute, and the same slope indicates a similar source of water. A Schoeller diagram allows multiple main ionic components to be represented in the same diagram to analyze the concentration changes and hydrochemical trends of the main ions in water samples [27]. If a straight line joining the points for two elements in one type of water is parallel to another straight line joining the parts for the same elements in another type of water, the ratio of these elements is the same in either case. If the waters have different concentrations, they will appear on the graph one above the other. Thus, indicating the relative movement of the groundwater between the points of origin of these analyses [9].

Water sample's solute concentrations in 1995 and 2016 were plotted separately into a Schoeller diagram, the results of which are shown in Figures 6 and 7. It intuitively showed the correlation between solutes and each other, and the fact that the slope was similar to the concentration line indicated that the source of the groundwater was similar.

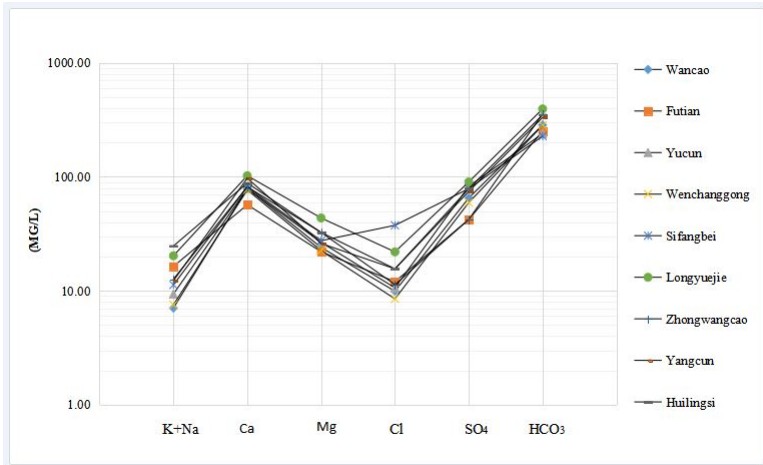

**Figure 6.** Schoeller diagram of the study area in 1995.

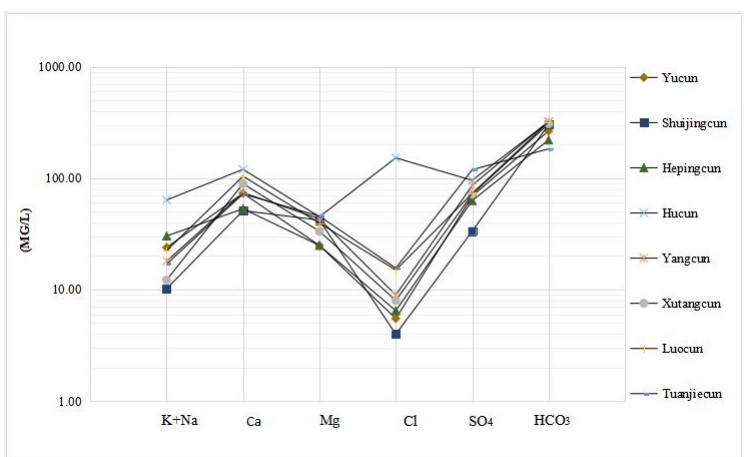

**Figure 7.** Schoeller diagram of the study area in 2016.

According to Schoeller's results, the changing trend of the curve of water samples in the study area was generally similar, indicating that groundwater sources in the study area were the same. In the diagram, the slope of solute concentration in Hucun was different from others in the whole study area. The concentration of Cl$^{-}$ was significantly higher. The chemical components of groundwater in Hucun were the result of pollution as the water sample points were near the septic tank during the sampling proceeded.

### 3.2. Groundwater Quality Assessment Results

The correlation degree of groundwater at eight sampling points in the study area was obtained from the expression of the correlation degree (1) and (2).

Taking the sample point S1 (Yucun) as an example, the concentration of chlorine ion, ammonia, nitrogen, and iron ion were lower than the limitation of rank I. The total hardness, the concentration of total soluble solids and sulfate ion were between the limitation of rank I and II. The concentration of nitrate ion was between the limitation of rank III and IV, and the concentration of fluorine concentration was higher than rank IV. Then, the contact number of S1 was calculated by:

$$\mu1(\text{Yucun}) = \frac{3}{8} + \frac{3}{8}i + \frac{0}{8}j + \frac{1}{8}k + \frac{1}{8}l$$

Similarly, the contact number of the other seven points were obtained:

$$\mu2(\text{Shuiingcun}) = \tfrac{4}{8} + \tfrac{2}{8}i + \tfrac{0}{8}j + \tfrac{1}{8}k + \tfrac{0}{8}l \quad \mu3(\text{Hepingcun}) = \tfrac{4}{8} + \tfrac{3}{8}i + \tfrac{0}{8}j + \tfrac{0}{8}k + \tfrac{1}{8}l$$

$$\mu4(\text{Hucun}) = \tfrac{3}{8} + \tfrac{1}{8}i + \tfrac{2}{8}j + \tfrac{1}{8}k + \tfrac{1}{8}l \quad \mu5(\text{Yangcun}) = \tfrac{4}{8} + \tfrac{2}{8}i + \tfrac{1}{8}j + \tfrac{0}{8}k + \tfrac{1}{8}l$$

$$\mu6(\text{Xutangcun}) = \tfrac{4}{8} + \tfrac{2}{8}i + \tfrac{1}{8}j + \tfrac{0}{8}k + \tfrac{1}{8}l \quad \mu7(\text{Tuanjicun}) = \tfrac{4}{8} + \tfrac{2}{8}i + \tfrac{1}{8}j + \tfrac{0}{8}k + \tfrac{1}{8}l$$

$$\mu8(\text{Luocun}) = \tfrac{4}{8} + \tfrac{1}{8}i + \tfrac{2}{8}j + \tfrac{0}{8}k + \tfrac{1}{8}l$$

By using Equations (3) and (4), the correlation degree of each evaluation index of the groundwater was calculated, and the average correlation degree was obtained according to Equation (5).

$$\mu TH = 0.1 + 0.9i + 0j + 0k + 0l$$
$$\mu TDS = 0.68 + 0.32i + 0j + 0k + 0\,l$$
$$\mu SO_4^{2-} = 0.809 + 0.191i + 0j + 0k + 0l$$
$$\mu Cl^- = 1 + 0i + 0j + 0k + 0l$$
$$\mu NO_3^- = 0 + 0i + 0.7j + 0.3k + 0l$$
$$\mu F = 0 + 0i + 0j + 0k + 1l$$
$$\mu NH_4^+ = 1 + 0i + 0j + 0k + 0l$$
$$\mu Fe = 1 + 0i + 0j + 0k + 0l$$
$$\mu1 = 0.5736 + 0.1764i + 0.0875j + 0.0375k + 0.1250l$$

In the same way, the average correlations of eight samples were obtained. According to Equation (6), the compound water quality grade, $G_m$, of the groundwater was obtained. The correlation degree of the eight samples obtained was analyzed one by one. Then, the set pair posture of the degree of association was determined according to the state power table of the five-element connection number, as shown in Table 5.

**Table 5.** Average contact number of each point and its set pair situation.

| Sites | *a* | *b* | *c* | *d* | *e* | $G_m$ | IDC State | Set Pair Situation |
|---|---|---|---|---|---|---|---|---|
| Yucun | 0.5736 | 0.1764 | 0.0875 | 0.0375 | 0.1250 | III | $a > e, a > b, b > c, c > d, d < e$ | identity state power (3) |
| Shuijingcun | 0.6844 | 0.1438 | 0.0175 | 0.1075 | 0.0000 | II | $a > e, a > b, b > c, c < d, d > e. \, e = 0$ | identity state power (7) |
| Hepingcun | 0.7680 | 0.1070 | 0.0000 | 0.0000 | 0.1250 | II | $a > e, a > b, b > c, c < d, d > e$ | identity state power (6) |
| Hucun | 0.4430 | 0.2505 | 0.1596 | 0.0219 | 0.1250 | III | $a > e, a < b, b > c, c > d, d < e$ | identity state power (49) |
| Yangcun | 0.6418 | 0.1808 | 0.0525 | 0.0000 | 0.1250 | II | $a > e, a > b, b > c, c > d, d < e$ | identity state power (3) |
| Xutangcun | 0.6366 | 0.1884 | 0.0500 | 0.0000 | 0.1250 | II | $a > e, a > b, b > c, c > d, d < e$ | identity state power (3) |
| Tuanjicun | 0.5965 | 0.1593 | 0.1193 | 0.0000 | 0.1250 | II | $a > e, a > b, b > c, c > d, d < e$ | identity state power (3) |
| Luocun | 0.5425 | 0.2758 | 0.0567 | 0.0000 | 0.1250 | III | $a > e, a > b, b > c, c > d, d < e$ | identity state power (3) |

The results showed that the quality evaluations of groundwater in Yucun, Hucun, and Luocun were in rank III, and the rest were in rank II. The situation for identity state power was 49 in Hucun, which means the quality of water was the worst.

Belonging to identity state power 3, with potential and further analysis by contact degree all belonging to the same rank, the groundwater quality condition was different for $\mu5 > \mu6 > \mu7 > \mu8$, and the water quality varied from good to bad in the order: Yangcun, Xutangcun, Tuanjiecun, and Luocun. For the value of $\mu5 > \mu6 > \mu7 > \mu1 > \mu8$, the quality of the groundwater of Yucun was worse than that of Yangcun, Xutangcun, and Tuanjiecun. The poor water quality of Yucun was affected by human activities. The quality of the groundwater of Shuijingcun and Hepingcun, located at the top of the fan, was good.

## 4. Conclusions

The five-element connect number SPA method was applied for the evaluation of impacts of groundwater quality assessment in the Gaoqiao diluvial fan. The chemical composition of the shallow groundwater reflected the conditions of the hydrogeological condition and was also affected by human activities. The sources of groundwater were the same and the groundwater quality in the study area was more in rank II. The basic hydrochemical types of shallow groundwater were $HCO_3$-Ca·Mg and

$HCO_3 \cdot SO_4$-Ca·Mg. The quality of groundwater in the study area changed slightly from the top to the edge of the fan, except for Yuncun and Hucun which was influenced by human activities. The assessment result can provide a scientific basis for the groundwater environmental assessment of changing process and its influencing factors affected by the development of tourism and increase of visitors in the hydrogeological unit of the Gaoqiao diluvial fan.

**Author Contributions:** Y.Y. and Q.C. performed the field investigation, W.X. conceived and designed the experiments; Y.Y. and J.C. performed the data analysis and wrote the paper.

**Funding:** This research was funded by the National Natural Science Foundation of China (NO. 41202213), Scientific Research Fund of Sichuan Provincial Education Department (NO. 18ZA0060), China Postdoctoral Science Foundation (2017M610598, 2018T110963), "Training program for young and middle-aged key teachers" supported by Chengdu University of Technology, Sichuan province, China. We received above funds for covering the costs to publish in open access.

**Acknowledgments:** The authors acknowledge the constructive and detailed comments of the reviewers and editors to improve this paper.

**Conflicts of Interest:** The authors declare no conflict of interest.

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
