# Peer review of "Hydrochemical Characteristics and Groundwater Quality Assessment in the Diluvial Fan of Gaoqiao, Emei Mountain, China"

_sustainability, doi:10.3390/su10124507_

Round 1
Reviewer 1 Report
- Introduction is too general with too many unnecessary listing of previous works/papers. Authors did not make a real introduction in their topics.
- English is weak and sometimes it is very difficult to understand the real meaning of the sentences.
- In the chapter Study area, authors provided very limited information on local geology and hydrogeology (particularly on petrographic and mineralogical properties of the rocks in the area under investigation).
- The location map and detail hydrogeological map of the area (containing description of hydrogeological units) are missing. Pollution sources were not indicated on map(s). The scale on the map on Fig1 is not visible.
- There are no explanation on characteristics of sampling points (observation wells, springs or?)
- Authors did not explain the sampling campaign in 1995 (as a reference to their work in 2016).
- Major remarks is that authors interpreted groundwater quality taking into account only major cations and anions and indicators. They did not provide measurements on other potentially important chemical parameters: phosphate (it should be relevant for this area), heavy metals, organic substances - at least total pesticides etc.
- Description of the methods used is weak and unclear, particularly set pair analysis method.
- Authors refer to 1995 campaign as a background value. Background value has a different meaning in hydrogeology, meaning concentration free of human impact.
- The presentation of results is weak: figures in table are inconsistent – e.g. different number of decimal places in table 2; too many repetition of numbers in text on page 8; low quality Schoeller diagram (see fig. 3) – 1995 and 2016 data should be presented separately, low quality Piper diagram on Fig 4. (See text crossing the figure) etc.
- Conclusions are premature. There are too many assumptions, without real evidence from results.
Author Response
- Introduction is too general with too many unnecessary listing of previous works/papers. Authors did not make a real introduction in their topics.
Reply: The introduction has been revised to highlight the research on the assessment method of groundwater quality influenced by human activities.
English is weak and sometimes it is very difficult to understand the real meaning of the sentences.
Reply: English has been checked and much improved to make it easier to understand.
- In the chapter Study area, authors provided very limited information on local geology and hydrogeology (particularly on petrographic and mineralogical properties of the rocks in the area under investigation).
Reply: Local geology and hydrogeology information has been supplemented in the “study area”.
The accumulation area of Gaoqiao diluvial fan to the NW-SE, covers an area of about 16 km2, longitudinal length about 7km. The stratum is quaternary (Q4, Q33, Q32, Q31, etc.) with wide horizontal distribution. The rich groundwater lateral runoff in the mountainous area supplies the diluvial fan, and the groundwater discharge patterns are wells and artesian springs.
The deposits in the diluvial fan are mainly basalt, sandstone and dolomite, with intermixed filling of sand and clay. The thickness of accumulation layer is large and the separation is poor. Diluvial fan from top to edge is divided into three phases. In the top phase, the gravel particle size is large, about 2-5 cm in diameter, and the water permeability is strong. In the middle phase, the component material are smaller than the top phase, mainly composed of gravel, sand and silt. In the edge phase, the composition of the material is small, composed of sub - sandy soil, sub - clay composition. Schematic diagram of diluvial fan section is shown in the figure.
The location map and detail hydrogeological map of the area (containing description of hydrogeological units) are missing. Pollution sources were not indicated on map(s). The scale on the map on Fig1 is not visible.
Reply: The location map and detail hydrogeological map has been supplemented and modified in the Figure1,Figure2 and Figure3 according to the hydrogeological conditions of the fan. Pollution sources and the scale were indicated on the map of Figure 2.
There are no explanation on characteristics of sampling points (observation wells, springs or?)
Reply: Characteristics of the sampling points have been supplemented in the table 1.
Table 1 characteristics of sampling points
Time | Geographic Location | Landform Location | Characteristic |
1995 | Wancao | Top | Spring |
Futian | Top | Spring | |
Yucun | Top | Well | |
Wenchanggong | Top | Spring | |
Sifangbei | Top | Well | |
Longyuejie | Middle | Well | |
Zhongwangcao | Middle | Well | |
Yangcun | Middle | Well | |
Huilingsi | Edge | Well | |
2016 | Yucun | Top | Spring |
Shuijincun | Top | Well | |
Hepingcun | Top | Well | |
Hucun | Middle | Well | |
Yangcun | Middle | Well | |
Xutangcun | Edge | Spring | |
Tuanjiecun | Edge | Well | |
Luocun | Edge | Well |
- Authors did not explain the sampling campaign in 1995 (as a reference to their work in 2016).
Reply: The sampling campaign in 1995 was not explained in the reference of Wuyong. But from the water quality evaluation aim of the study in 1995, the distribution of samples in the fan took the groundwater flow direction into consideration.
Major remarks is that authors interpreted groundwater quality taking into account only major cations and anions and indicators. They did not provide measurements on other potentially important chemical parameters: phosphate (it should be relevant for this area), heavy metals, organic substances - at least total pesticides etc.
Reply: The typical endemic disease in study area was the "dental fluorosis", showing that the groundwater was polluted by F. Therefore, F was evaluated and analyzed specially as the typical pollution from human activities.
Description of the methods used is weak and unclear, particularly set pair analysis method.
Reply: The introduction of set pair analysis method (SPA) was added so that readers can understand it more easily.
Authors refer to 1995 campaign as a background value. Background value has a different meaning in hydrogeology, meaning concentration free of human impact.
Reply: In this paper, the chemical parameters in 1995 was taken as “reference value” in the comparative analysis, not “as a background value”. It has been modified.
The presentation of results is weak: figures in table are inconsistent – e.g. different number of decimal places in table 2; too many repetition of numbers in text on page 8; low quality Schoeller diagram (see fig. 3) – 1995 and 2016 data should be presented separately, low quality Piper diagram on Fig 4. (See text crossing the figure) etc.
Reply: These problems have been modified and the figures have been improved.
Conclusions are premature. There are too many assumptions, without real evidence from results.
Reply: The conclusions have been modified and improved according with the result.
Reviewer 2 Report
The quality of groundwater is a key issue that relates to water security and human
health. To understand the quality of groundwater in diluvial fan of Gaoqiao, Emei
Mountain in China, the authors applied descriptive statistics, Piper diagram, Schoeller
diagram and some other approaches to analyze the hydro-chemical characteristics of
the groundwater and its influencing factors in the study area. Then the authors
adopted the Set Pair Analysis (SPA) method to assess the quality of shallow
groundwater in the study area based on the experimental analysis results. Their results
show that the Ca2+ and HCO3- are the main cation and anion in the groundwater for
the study area respectively and the ion concentration is closely related to the
dissolution of sediments. The overall groundwater quality was ranked as III level
(Medium quality). The value of this study lies on possibly providing some scientific
basis for the development and utilization of water resources and the improvement of
water security. However, the novelty of the study was not accentuated and some
methods were not clearly described.
General Comments:
1. What I most concern about the manuscript is its novelty. Generally speaking, the
topic of this study is to assess the quality of groundwater at a diluvial fan region
and all the methods applied here (descriptive statistics, Piper diagram, Schoeller
diagram, SPA) are from previous studies. Neither the topic nor the method is new.
Therefore I recommend the authors to fully stress what is new in the study
(compared with previous works) and what new knowledge we can learn from their
results.
2. The SPA method is a key part in the assessment of the groundwater quality.
However, throughout the section 2.3, the SPA method was badly introduced,
making readers difficult to understand the approach. For example, what is the
statistical meaning for the connection degree? It is difficult for me to understand
that how the can we judge the relationship between A and B from equations (1) to
(5). The meaning for the “identity”, “discrepancy”, “contrary” degrees are also
unclear. Generally speaking, the section 2.3 may need to be rewritten to make the
readers who are unfamiliar to the SPA method can understand the results as well.
Specific Comments:
L14-15: the statement of “the influence of human activities on groundwater” is a
much larger topic than what the manuscript described. This study is more like an
assessment for the groundwater quality than a study for influence of human beings.
L46-54: These references are not closely related to this study and the paragraph could
be simplified.
L67-68:
What are the relationships between the “revealing influencing factors”
(described in last paragraph) and the “groundwater quality evaluation”
(described in this paragraph)? Every paragraph should be logically
linked with each other.
L85: Replace “set pair analysis” by SPA
L97: Replace “of the rational development” by “for the rational development”.
L136: Replace “of regional pollution sources” by “to regional pollution sources”.
L165: “ive-element” should be “five element”.
L295-296:
Why can we get a conclusion here that “the Cl- in the study area is
caused by artificial sources” from “its poor correlation with salt
dissolution”? More explanation is needed.
L366: Replace “Comprehensive analysis” by “By comprehensive analysis”.
Fig.3: The figure is not complete. Some labels are missing.
Fig.4: There are some overlaps. The figure needs be reedited.
Fig.5: The symbols of the cations and anions are incomplete (e.g. Ca should be Ca2+)
Table.5:
More introduction for the index of “set pair situation” should be added
in the main text to let readers understand the results more easily.
Author Response
The quality of groundwater is a key issue that relates to water security and human
health. To understand the quality of groundwater in diluvial fan of Gaoqiao, Emei
Mountain in China, the authors applied descriptive statistics, Piper diagram, Schoeller
diagram and some other approaches to analyze the hydro-chemical characteristics of
the groundwater and its influencing factors in the study area. Then the authors
adopted the Set Pair Analysis (SPA) method to assess the quality of shallow
groundwater in the study area based on the experimental analysis results. Their results
show that the Ca2+ and HCO3- are the main cation and anion in the groundwater for
the study area respectively and the ion concentration is closely related to the
dissolution of sediments. The overall groundwater quality was ranked as III level
(Medium quality). The value of this study lies on possibly providing some scientific
basis for the development and utilization of water resources and the improvement of
water security. However, the novelty of the study was not accentuated and some
methods were not clearly described.
Reply: On the basis of hydrogeochemical components analysis in 2016 compared with in 1995, the five-element connection number SPA method was proposed and applied to evaluate the groundwater quality of the Gaoqiao alluvial-diluvial fan under the influence of human activities in this work. The assessment result can provide scientific basis for the changing process and its influencing factors assessment of groundwater and sustainable utilization of the groundwater resources in the hydrogeological unit of Gaoqiao alluvial-diluvial fan.
General Comments:
1. What I most concern about the manuscript is its novelty. Generally speaking, the
topic of this study is to assess the quality of groundwater at a diluvial fan region
and all the methods applied here (descriptive statistics, Piper diagram, Schoeller
diagram, SPA) are from previous studies. Neither the topic nor the method is new. Therefore I recommend the authors to fully stress what is new in the study (compared with previous works) and what new knowledge we can learn from their results.
Reply: This paper focus on the evaluation of the current situation of groundwater quality influenced by the human activities and its recovery degree of the Gaoqiao alluvial-diluvial fan based on the hydrogeochemical components analysis in 2016 compared with 1995, based on the five-element connection number SPA method.
The five-element connect number SPA method is superior to the uncertainties since it is not necessary to distinguish the information definitely and it has been applied in evaluation of impacts of groundwater quality assessment. The chemical composition of the shallow groundwater reflected the conditions of the hydrogeological condition and it was also affected by human activities in the Gaoqiao alluvial-diluvial fan. The quality of groundwater in the study area changed from the top to the edge of the fan, and the quality of groundwater was getting worse and worse except for Yuncun and Hucun influence by human activities. The assessment result can provide scientific basis for the changing process and its influencing factors assessment of groundwater and sustainable utilization of the groundwater resources in the hydrogeological unit of Gaoqiao alluvial-diluvial fan.
2. The SPA method is a key part in the assessment of the groundwater quality.
However, throughout the section 2.3, the SPA method was badly introduced,
making readers difficult to understand the approach. For example, what is the
statistical meaning for the connection degree? It is difficult for me to understand
that how the can we judge the relationship between A and B from equations (1) to
(5). The meaning for the “identity”, “discrepancy”, “contrary” degrees are also
unclear. Generally speaking, the section 2.3 may need to be rewritten to make the
readers who are unfamiliar to the SPA method can understand the results as well.
Reply: The SPA method is described in detail in the text and the section 2.3 has be rewritten to make the readers understand the SPA method easily.
Set pair analysis is a theory to study the problem of determination uncertainty. Its core idea is to regard the problem of determination uncertainty as a system and analyze the relations and changes between things from three aspects: identity (same), discrepancy (different) and contrary (opposite). The core theory of set pair analysis is as follows: firstly, set pairs of two relative sets in uncertain systems are established. Then, using identity, discrepancy and contrary to analyze and calculate attributes. Finally, the degree of connection of a pair of combinations can be established according to the "identity ", "discrepancy "and "contrary ". Set Pair Analysis (SPA) is a modified uncertainty theory considering both certainties and uncertainties as an integrated certain–uncertain system and analyzing things from three aspects such as identity, discrepancy and contrary. In the SPA identity, discrepancy and contrary are transformed under certain conditions, and the evaluation of groundwater environmental quality can be realized through this relationship between interconnection and restriction.
Specific Comments:
L14-15: the statement of “the influence of human activities on groundwater” is a
much larger topic than what the manuscript described. This study is more like an
assessment for the groundwater quality than a study for influence of human beings.
Reply: The influence of human activities on the groundwater has been supplemented in the “study area” and the relationship with the groundwater quality has been analyzed in the “Result”. The abstract is modified to highlight the topic of the article.
L46-54: These references are not closely related to this study and the paragraph could
be simplified.
Reply: The main relevant references about the five-element connect number set pair analysis have been supplemented and the others have been modified and simplified.
L67-68: What are the relationships between the “revealing influencing factors” (described in last paragraph) and the “groundwater quality evaluation” (described in this paragraph)? Every paragraph should be logically linked with each other.
Reply: The relationships between the human activities with the groundwater quality have been analyzed in the “Result”.
L85: Replace “set pair analysis” by SPA.
Reply: It has been modified.
L97: Replace “of the rational development” by “for the rational development”.
Reply: This part has been modified.
L136: Replace “of regional pollution sources” by “to regional pollution sources”.
Reply: This part has been modified.
L165: “Five-element” should be “five element”.
Reply: It has been modified.
L295-296: Why can we get a conclusion here that “the Cl- in the study area is caused by artificial sources” from “its poor correlation with salt dissolution”? More explanation is needed.
Reply: The conclusion of “the Cl- in the study area is caused by artificial sources” from “its poor correlation with salt dissolution” has been modified based on further analysis. The groundwater quality was influenced more heavily by the human activities than the natural water-rock interaction.
L366: Replace “Comprehensive analysis” by “By comprehensive analysis”.
Reply: It has been modified.
Fig.3: The figure is not complete. Some labels are missing.
Reply: It has been modified.
Fig.4: There are some overlaps. The figure needs be reedited.
Reply: It has been modified.
Fig.5: The symbols of the cations and anions are incomplete (e.g. Ca should be Ca2+)
Reply: The figure 5 has been deleted according to the needs of research.
Table.5: More introduction for the index of “set pair situation” should be added in the main text to let readers understand the results more easily.
Reply: The main relevant references about the five-element connect number set pair analysis have been supplemented in the introduction of the paper. The methods are described in detail in the text.
SPA has been applied in evaluation of impacts of groundwater quality assessment [6], water resources management problems [15-17] and other groundwater analysis issues.
6. Zhu H, Ren X, Liu Z. A new four-step hierarchy method for combined assessment of groundwater quality and pollution. [J]. Environmental Monitoring & Assessment, 2018, 190(1):50.
15. Pan Z, Wang Y, Jin J, et al. Set pair analysis method for coordination evaluation in water resources utilizing conflict [J]. Physics & Chemistry of the Earth Parts A/b/c, 2017, 101.
16. Chen L Y,Fu Q, et al. Application of Five-Element Connection Number to the Quality Assessment of Eutrophication in Lakes[J]. Research of Environmental Sciences, 2008, 21(3):82-86.
17. Yu F, Qu J, Li Z, et al. Application of set pair analysis based on the improved five-element connectivity in the evaluation of groundwater quality in XuChang, Henan Province, China[J]. Water Science & Technology Water Supply, 2017, 17(3):632-642.
Reviewer 3 Report
Authors presented the assessment of groundwater quality using Set Pair Analysis (SPA). It would be interesting to readers of this journal and is important to manage groundwater resources. However, I have several suggestions before publication of this manuscript.
Authors said that the quality of groundwater were getting worse and worse. Do you mean that the quality of groundwater is worse in 2016 than in 1995? But, authors sampled groundwater at different points between 1995 and 2016. In addition, it seems that groundwater quality is not significantly changed from Table 3, Piper diagram, and Schoeller diagram. The slightly increase of ions can be attributed to the water-rock interaction along flow path according to the mineralogy of aquifers. In addition, it seems that some pollution indicators (F and Cl) are originated from point sources (F in mine at Yucun, Cl in septic tank at Hucun). Authors also said that chemical composition of the shallow groundwater reflected the hydrological condition. But I did not see the related discussion in the manuscript.
Could you compare the results of rank on groundwater quality according to between groundwater quality standards of China and SPA? Can we see the usefulness of SPA from this comparison?
Page 2, line 80-82. I don’t understand why SPA has obvious advantages in dealing with fuzzy, random, etc. Could you supply some references?
Page 8, line 221-223. What’s the meaning of basis cation and anion?
Page 9, Figure 4 and Figure 5. Please insert the legend for samples.
Page 9, line 241-242. I can’t confirm that the increase in TDS results from human activities of tourism.
Page 10, line 274-276. I wonder that we can say that sources of groundwater is similar from the result of Schoeller diagram without oxygen and hydrogen isotopes of water?
Author Response
Could you compare the results of rank on groundwater quality according to between groundwater quality standards of China and SPA? Can we see the usefulness of SPA from this comparison?
Reply:
(1) The groundwater points were sampled and analyzed at similar area of the fan between 1995 and 2016, showing in the figure 2.
(2) “The quality of groundwater were getting worse and worse” has been modified to “The quality of groundwater changed slightly” according to the increase of ions of the shallow groundwater.
(3) It did not mean that the quality of groundwater is worse in 2016 than in 1995. The difference of quality of groundwater between them was discussed aiming to find out the reason of change.
(4) The relationship of the chemical composition of the shallow groundwater with the hydrological condition was supplemented in the “Result and discussion”.
Page 2, line 80-82. I don’t understand why SPA has obvious advantages in dealing with fuzzy, random, etc. Could you supply some references?
Reply:These references have been supplemented.
4. Wang, W., J, J., Ding, J. et al. A new approach to water resources system assessment—set pair analysis method [J]. Sci. China Ser. E-Technol. Sci. (2009) 52(10): 3017-3023.
14. Wang Y , Jing H , Yu L , et al. Set pair analysis for risk assessment of water inrush in karst tunnels[J]. Bulletin of Engineering Geology and the Environment, 2016.
15. Wang Y , Yang W , Li M , et al. Risk assessment of floor water inrush in coal mines based on secondary fuzzy comprehensive evaluation[J]. International Journal of Rock Mechanics and Mining Sciences, 2012, 52(none):50-55.
16.Yue W , Cai Y , Rong Q , et al. A hybrid life-cycle and fuzzy-set-pair analyses approach for comprehensively evaluating impacts of industrial wastewater under uncertainty[J]. Journal of Cleaner Production, 2014, 80(7):57-68.
33. Wang X, Wang W S, Ding J. Set pair analysis and its application to hydrology and water resources [J]. Adv Sci Tech Water Resour, 2006, 26(4): 9-11.
34. Wang D, Zhu Y S, Zhao K Q. Research and application of model based on set pair analysis and fuzzy set theory for assessment of water eutrophication [J]. J Hydrol, 2004, 24(3): 9-13
Page 8, line 221-223. What’s the meaning of basis cation and anion?
Reply:“Basis anion and cation” has been modified to “basic cation and anion”, and basic cation refers to the chemical composition of K+, Na+, Ca2+ ,Mg2+, and basic anion refers to the chemical composition of HCO3-, Cl-, SO42- , CO32-.
Page 9, Figure 4 and Figure 5. Please insert the legend for samples.
Reply:The legend for samples has been inserted.
Page 9, line 241-242. I can’t confirm that the increase in TDS results from human activities of tourism.
Reply:
“The growth of TDS was mainly caused by the increasing human activities of tourism” has been modified to “The slightly increase of TDS ions can be attributed to the water-rock interaction along the flow path according to the mineralogy of aquifers”.
Page 10, line 274-276. I wonder that we can say that sources of groundwater is similar from the result of Schoeller diagram without oxygen and hydrogen isotopes of water?
Reply:
The groundwater has similar source from the top of the fan and has similar water-rock interaction along the flow path according to the hydrological condition of the alluvial-diluvial fan.
Reviewer 4 Report
Please read and cite:
The
subject of the paper is interesting, due to the evaluation of groundwater quality, thus results can be of interest to a
large range of readers.
As follow, I suggest to read some paper to
the author.
Please in introduction, read and cite: Miglietta, P. P., Toma, P., Fanizzi, Bagordo, F., Migoni, D., Grassi, T., Serio, F., Miglietta, P. P., Lamastra, These paper can help you with other approaches monitoring groundwater quality and pollution.
F. P., De Donno, A., Coluccia, B., Migoni, D., ... & Serio, F.
(2017). A Grey water footprint assessment of groundwater chemical
pollution: case study in Salento (southern Italy). Sustainability, 9(5), 799.
Serio, F., Idolo, A., Guido, M., ... & De Donno, A. (2016). Using
the DPSIR framework to identify factors influencing the quality of
groundwater in Grecìa Salentina (Puglia, Italy). Rendiconti Lincei, 27(1), 113-125.
L., Ficocelli, S., Intini, F., De Leo, F., & De Donno, A. (2018).
Groundwater nitrate contamination and agricultural land use: A grey
water footprint perspective in Southern Apulia Region (Italy). Science of the Total Environment, 645, 1425-1431.
Author Response
Reply:
These papers had been read and cited in the introduction and the reference had been improved.
The “grey water footprint” (GWF) was used to assess the sustainability of pollution produce by human activities and to measure the amount of water required to assimilate a polluting load produced from anthropic activity(Serio F. 2018). The Driver-Pressure-State-Impact-Response (DPSIR) framework was providing a simplified description of the various components of a complex environmental system to obtain information useful for the management of water resources(Bagordo 2016), applied in a central area of the Salento peninsula to identify environmental and human factors influencing the quality of groundwater(Miglietta 2017)
References:
7. Serio, F., Miglietta, P. P., Lamastra, L., Ficocelli, S., Intini, F., De Leo, F., & De Donno, A. (2018). Groundwater nitrate contamination and agricultural land use: A grey water footprint perspective in Southern Apulia Region (Italy). Science of the Total Environment, 645, 1425-1431.
8. Miglietta, P. P. , Toma, P. , Fanizzi, F. P. , Donno, A. D., et al. (2017). A grey water footprint assessment of groundwater chemical pollution: case study in salento (southern italy). Sustainability, 9(5), 799.
9. Bagordo F , Migoni D , Grassi T , et al. 2016. Using the DPSIR framework to identify factors influencing the quality of groundwater in Grecìa Salentina (Puglia, Italy). Rendiconti Lincei, 27(1):113-125.